# Perceptions of the Impact of Artificial Intelligence among Internal Medicine Physicians as a Step in Social Responsibility Implementation: A Cross-Sectional Study

**DOI:** 10.3390/healthcare12151502

**Published:** 2024-07-29

**Authors:** Luminița-Mihaela Dumitrașcu, Delia-Andreea Lespezeanu, Corina-Aurelia Zugravu, Ciprian Constantin

**Affiliations:** 1Department of Accounting and Audit, Faculty of Accounting and Management Information Systems, The Bucharest University of Economic Studies, 010374 Bucharest, Romania; 2Doctoral School, Faculty of Medicine, “Titu Maiorescu” University, 031593 Bucharest, Romania; delia.lespezeanu@gmail.com; 3“Ion Pavel” Diabetes Center, National Institute of Diabetes, Nutrition and Metabolic Diseases “Prof. Dr. N.C. Paulescu”, 030167 Bucharest, Romania; 4Department of Nutrition, Hygiene and Ecology, Faculty of Midwifery and Nursing, “Carol Davila” University of Medicine and Pharmacy, 050474 Bucharest, Romania; corina.zugravu@umf.ro; 5The National Institute of Public Health, National Center of Risk Monitoring for Community, 050463 Bucharest, Romania; 6“Carol Davila” Central Military Emergency University Hospital, 010825 Bucharest, Romania; ciprian_constantin@yahoo.com; 7Research Metabolism Center, 010825 Bucharest, Romania; 8Faculty of Medicine, “Titu Maiorescu” University, 031593 Bucharest, Romania

**Keywords:** Artificial Intelligence, social responsibility, health sector, medical education, ethics

## Abstract

Artificial Intelligence (AI) has emerged as an essential tool in healthcare for optimizing healthcare delivery and improving patient outcomes. This study is motivated by using AI in healthcare as a step for social responsibility implementation. The research aimed to investigate the attitudes of healthcare professionals on this issue, and it assessed physicians’ opinions regarding their perceptions of AI and their intention to use and implement AI tools in their activity. An electronic survey was proposed during February–June 2024 to a sample of healthcare professionals (309 were admitted into the study, 62 males and 247 females, with a mean age of 42). The results of the survey highlighted both groups’ excellent perceptions of AI and the low perceived knowledge of AI, which arises from more technical questions. The use of AI in healthcare represents a step for social responsibility implementation; it is an unstoppable process, and stakeholders should take into consideration investing more in monitoring and training activities.

## 1. Introduction

Artificial Intelligence (AI) represents the ability of machines to perform different tasks that usually require human intelligence.

In healthcare, in the era of technology, many inventions have expanded the limits of medical treatment and diagnosis beyond the current capabilities. AI can improve the medical act and different technologies already in practice and is essential for managerial activities; it can improve diagnosis, treatment, administrative tasks, drug development, medical research, and education [1,2]. 

On 18 January 2024, the World Health Organization [1] released new guidance on the ethics and governance of AI for health to assist states in mapping the challenges and benefits of using AI. Among the benefits and risks of using it in the health sector, they mention

Diagnosis and clinical care, especially in radiology, imaging, and oncology. AI helps physicians to recognize at-risk patients. The main associated risk refers to inaccurate or incomplete diagnosis;Patient-centered applications are used to take medicines, improve diet, and engage in physical activity through chatbots, health monitoring, and risk prediction tools. The associated risks refer to inaccurate or incomplete statements, data privacy concerns, and low interaction with clinicians;Administrative tasks and AI tools help health professionals in their daily activities. The main risks that arise could be inaccuracies or generating a completely different response;Medical education. The main risks refer to the professional judgment that could be affected;Medical research and drug development could be extended. The main risks refer to the lack of accountability.

WHO issued a Regulation on AI for health on 19 October 2023. The main recommendations they made referred to Governments that should invest in proper infrastructure, laws, and rules to ensure an ethical implementation and to developers that should involve different stakeholders to execute defined tasks to promote patient welfare (https://www.globalcompliancenews.com/2024/02/17/https-insightplus-bakermckenzie-com-bm-healthcare-life-sciences-singapore-world-health-organization-releases-ai-ethics-and-governance-guidance-for-large-multimodal-models_01312024/, accessed on 8 February 2024) [3]. 

AI is gradually increasing its impact on everyday life in almost all sectors of activity and is becoming increasingly common. In medicine, it is of growing interest and is used to improve drug discovery, diagnosis, and decisional processes [4,5,6]. 

One of the main objectives of AI is to improve time management and efficiency in terms of performance, minimize costs, and satisfy customers. The rapid evolution of AI in the health sector creates numerous concerns and challenges. Although it is well known that the implementation of AI is over different sectors of activity, the potential of AI to drive revenues and increase profitability has opened more opportunities across the healthcare sector.

It is constructive to understand initiatives developed worldwide. Like a desirable solution, the European Union (EU) educates people to develop healthy lifestyles. It focuses on providing citizens secure access to personal health across EU borders and to personal medicine, encouraging patients to take care of themselves and stimulating interactions between patients and providers.

The research questions for the current research are as follows:(1)What are the perceptions and attitudes of physicians related to AI implementation in the health sector?(2)Could AI in the health sector represent a step in implementing social responsibility?

The paper focuses on Artificial Intelligence in medicine. Our findings contribute to the body of knowledge and aim to deepen the understanding of physicians’ perceptions related to AI’s implementation in the healthcare sector. The findings could provide a guideline to higher institutions on artificial intelligence and social responsibility areas to be incorporated into their syllabus. This will ensure that students, and future physicians, are equipped with abilities and skills that meet the current market demands. To the best of the authors’ knowledge, this is the first study to investigate physicians’ perspectives on artificial intelligence in our country. 

## 2. AI in the Health Sector and Social Responsibility

AI implementation in health aims to foster organizational changes in all departments. After reviewing relevant research studies, we found that, nowadays, AI offers organizations flexible and interactive alternatives to developing a model for the implementation of AI in business. 

AI algorithms were used to make treatment recommendations. However, the recommendations may conflict with the physicians’ ethical obligation to act in the best interest of patients [7] (Figure 1).

Social responsibility, accountability, and sustainability bring risk reduction benefits, make patients comfortable and transparent, and are essentials for AI implementation in the health sector [7]. AI should focus on ethical concerns (privacy, reducing harmfulness), risk mitigation (transparency, cooperation, reconfiguration), and skills (algorithm, data acquisition). 

The limits are represented by the fact that the gold standard for decision-making should be for physicians. From here arises another question related to the responsibility in case of an error. The results need to be corrected in case of poor data entry. It is not easy to compare results from different product providers since different techniques are used [8].

According to a report by Eurostat, only 8% of companies from the European Union use AI. Romania has the lowest implementation ratio, alongside Bulgaria, Greece, Hungary, and Poland. Only 1.5% of companies are using AI in Romania, compared with 15% in Denmark (https://ec.europa.eu/eurostat/databrowser/view/ISOC_EB_AI__custom_5458102/bookmark/table?lang=en&bookmarkId=3e222205-9074-4fb5-a2b3-9f31f96e7907, accessed on 22 June 2024) [9]. The AI Liability Directive is an initiative proposed by the European Commission for compensation related to harm and negligence in using AI. This regulation will increase ethics and transparency. The Authority for Digitalization of Romania and the Ministry of Research, Innovation, and Digitalization, through The Scientific and Ethical Council on AI, are the authorities that regulate AI. Other ministries are also involved and cooperate, such as the Ministry of Internal Affairs, Ministry of Finance, Ministry of Foreign Affairs, and Ministry of Education. At the current moment, there is no mandatory specific regulation in terms of AI in Romania. Shortly, the Government will enforce the AI Act, which was implemented at the EU level to prevent harm due to AI use. However, some laws may apply to different sectors of activity like finance, manufacturing, autonomous vehicles, and healthcare. Romania is typically aligned with EU rules and regulations. For example, in processing the personal data of individuals, Romania enforces privacy and data protection regulations, such as the General Data Protection Regulation (GDPR) through Law 190/2018. The competent authority is represented by The National Supervisory Authority for Personal Data Processing. Besides this, Romania, as a member state of the EU, has been required to implement EU 2022/2555 since January 2023, with a period of 21 months to transpose in the national legislative framework by September 2024. Moreover, the European Parliament approved the Cyber Resilience Act in March 2024 for cybersecurity assessment for digital products. Despite the lack of legislation, Romania took small steps in this direction and created Ion, which scans social media and public messages on an online platform to report to Government representatives. The National Strategy for AI 2024–2027, together with the methodological rules on its implementation, is another Romanian initiative to implement AI in public administration. Deep-fake Regulation 471/2023 is a pending draft addressing the responsible use of AI through video and/or audio content. The proposed penalties are between EUR 2000 and EUR 40,000. 

In the healthcare sector, Romania introduced diagnostic methods in hospitals (telemedicine platforms, Da Vinci robots for surgeries, and DeepcOS AIM for analyzing mammography images). 

## 3. Methodology of Research

The study’s primary purpose was to focus on AI in the health sector and to examine the perceptions of internal medicine physicians regarding AI implementation in the healthcare sector, as well as the main challenges and risks following social responsibility.

A web-based cross-sectional survey was performed between February and June 2024 among 309 physicians from Bucharest, Romania, from both the public and private sectors (Figure 2). A Google Forms survey was developed and distributed via the Research Metabolism Center from Bucharest. 

The participant selection process considered the physicians’ specialization. Each participant provided informed and voluntary consent before completing the survey under the ethical guidelines under the Declaration of Helsinki, accomplishing the transparency and ethical integrity of the current study. The Approval of the Ethic Committee is number 2, from 3 January 2024. 

According to the last published data from the National Institute of Statistics, Romania has 30,639 physicians, 21,430 stomatologists, 22,660 pharmacists, and 18,910 nurses. 

An electronic survey was proposed for a sample of healthcare professionals from February to June 2024. A total of 309 physicians were admitted to the study, of which 62 were males and 247 were females, with a mean age of 42 years (Figure 3).

Data were gathered through an in-depth questionnaire across hospitals and clinics in Romania from 309 health professionals, which enriched the understanding of the need for social responsibility actions and activities when integrating AI into medicine. The number of physicians included in the sample was 370, of which 60 were excluded because they still needed to complete the survey, and another one was due to the missing data in the completed survey. The findings show a continuous intention to use AI despite the lack of regulation on both AI and social responsibility in the health sector. We used open-ended questions, true or false, and multiple-choice questions to explore both concepts. A total of 309 questionnaires were collected in different rounds (Table 1).

The first stage of the study was exploratory questionnaires tested on a small sample (*n* = 10). In the second stage, starting from the answers received, we reevaluated and rephrased the questions. We used non-random or convenience sampling since it was easy to have access to data (geographical proximity, availability at a given time, and willingness to participate in the research). Another reason why we used convenience sampling is that it is often used in qualitative and medical research studies when it involves selecting participants that are available around a particular location. We are researching physician perceptions of AI implementation in healthcare in the city of Bucharest, Romania. We have determined that a sample of 309 physicians is sufficient to answer our research question and to discover the physicians’ perceptions, attitudes, and opinions on AI. The study can be developed in future research paths starting from the questionnaire developed in the current manuscript (Table 2).

To collect our data, we sent messages through a pre-existing group of internal medicine physicians and asked them if they wanted to participate in our research and to complete our online survey. We reached out to our respondents through different channels, sending the link through email and sending them gentle reminders to complete the survey or WhatsApp messages also as reminders. We used the social media platform where they were more active and ready to help us with their answers. Another important aspect is timing. We avoided posting it during busy times when they were maybe less likely to respond. To maximize the response ratio, we continued to invite them to complete the survey and sent them reminders on different days and times, from February to June 2024, with the shortened URL link, until the sample size was reached. We mentioned that we did not offer monetary or other incentives to complete our survey. As a method of research, we used the descriptive analysis method. The limit of our research is represented by the fact that it cannot be generalized to the target population.

We project some measures to be taken, following social responsibility, to implement AI within the health sector. This study is a step for future research to establish which actions and activities need to be taken to have a socially responsible AI implementation. 

In this regard, we provided a questionnaire to doctors, based on which we analyzed the professionals’ points of view regarding AI. The survey targeted specialist doctors and public and private sector residents (Table 3). Thus, people with experience in the field who could offer us a relevant opinion were targeted. 

The questionnaire consists of 17 questions (4 questions to identify the profile of the respondents, including age, gender, sector of activity, and professional experience) and 13 questions related to the studied issue (knowledge of AI, perception of its role in the medical system, training, and education, risks, potential tasks to be developed by AI, and the usefulness of AI) (Table 2). 

Most participants are women (80%) (Table 3). The respondents have an average age of 42 years and an average professional experience of up to 20 years (Figure 4).

The research results enrich our understanding of how socially responsible AI is in the health sector. Our findings clarify how ethics, risk mitigation, and other skills should be implemented to develop a socially responsible AI system. The impact on professionals is discussed, as well as the impact on improving the patient’s satisfaction and the organization’s market performance

## 4. Results and Discussion

The global AI market in the health sector was valued in 2024 at USD 20.9 billion, and it is estimated (Artificial Intelligence (AI) in Healthcare Market Size, Share, Industry Report, Statistics—2032 (marketsandmarkets.com)) to reach USD 148.4 billion; this growth has been driven by the large datasets from the health sector, the growing need to improve healthcare services, the need to reduce costs, and many other factors [10]. 

AI has the potential to ensure elderly care, contribute to the well-being of people with dementia and other chronic diseases, detect health issues, manage the medication and other treatment plans of patients, improve resource allocation, and optimize costs. However, AI also faces some impediments, such as data privacy, ethical issues, cybersecurity risks, high costs, and expertise barriers.

In November 2023, the World Health Organization (WHO) published regulatory guidelines for implementing artificial intelligence in healthcare. These guidelines address transparency, data privacy, data validation, risk management, data quality, and encouraging cooperation among stakeholders. Some solutions could be represented by standardization, partnerships between public and private institutions, proactive measures, and cooperation. 

In the current research, we examined the physicians’ perceptions about implementing AI. The results showed us that healthcare professionals are skilled and competent in other areas and open to AI training and education (Figure 5). 

Almost all respondents (96%) believe that AI will impact the health sector, but their answers are different. According to Figure 6, 69% believe that AI will improve the flow of activities, 24% said that it would have a revolutionary role, and the others affirmed that *the use of AI would represent both an advantage and a problem because robots cannot practice medicine*; *it will replace physicians in the relationship with the health insurance houses, with the related guidelines and protocols; it will help for better time management; it will help in the dimensioning of services, budgets, and services quality; it will increase the revenues of software developers and suppliers; it will replace the physicians; or it will confuse them*. In contrast, others did not observe any impact (Table 4).

The majority of the participating physicians said they would like to receive more training in artificial intelligence. A total of 92% of them said they have a positive attitude toward using artificial intelligence (Figure 7). Many physicians have declared the desire to use Artificial Intelligence tools in the future and have expressed the intention to participate in training programs. Some differences were observed, including age, gender, and experience. To the best of our knowledge, this is the first study from Romania regarding the attitudes toward artificial intelligence, as a step of social responsibility, and represents an added value to the current research in this field to guide the education and the practice within the health sector. 

When asked about their opinion on AI, the majority of physicians answered that AI will help them in their daily activities, reduce redundant activities, help in giving a second opinion, maintain the quality of health services, help with paperwork and diagnosis establishment, and help in statistical data processing, while other physicians have a negative perspective associated with AI, thinking that it will eliminate empathy.

The main applications of AI in healthcare include: Robot-assisted surgery is carried out by the Da Vinci robot, which is less invasive, and the Smart Tissue Autonomous Robot, which can perform surgeries with better precision. During this time, AI was involved in different surgeries, such as gynecological surgeries, prostatectomies, and the robotic surgery of cardiac valves [11,12].Administrative workflow (significant healthcare spending is due to redundant administrative tasks such as documentation, reviewing patient records, and managing patients’ files, which consume not only time but also effort and money).Fraud detection.Radiology and medical imaging.Clinical trials and research.Managing electronic health records.Disease diagnosis and clinical decision-making.Risk assessment AI can process real-time data to identify high-risk patients and predict disease evolution.Telemedicine and virtual health assistants are used to make recommendations and to improve access to healthcare services. Vital signs and symptoms can be tracked.With AI’s help, drug discovery can predict potential drug interactions.Patient monitoring and carePatient triage and resource allocationPatient prognosisTreatment decision-makingReducing medication errorsCybersecurity.

Risk, and especially cybersecurity risk, is one of the main concerns of any organization regardless of its sector of activity, and it also represents a concern for the majority of physicians from our sample (Figure 8). A cyberattack affects both the organization and its image. All companies are adopting AI to protect data and to detect strange behavior. AI has a significant impact on medicine. It can affect the treatment and diagnosis and also guide resource allocations. Other perceived risks refer to reducing the quality of the medical act, increasing the number of errors and malpractice cases, decreasing the skills of physicians, increasing dependence on a reporting instrument, the lack of empathy, and fear of job loss, while other physicians do not perceive any risks from AI implementation. From the physicians’ perspectives, there are many concerns regarding job displacement, reliability, and others. Addressing all these challenges requires many investments in training and focusing on cooperation between healthcare institutions and technology developers to obtain superior patient outcomes.

We were also interested in finding out the opinions of physicians regarding the use of AI within their daily activity, and 87% of them answered that they would accept using it (Figure 9).

Starting from here, another question of interest is the physicians’ perceptions of the degree of difficulty in implementing AI. Only 36% of them consider it easy (Figure 10). 

Another question that arises from here refers to potential tasks to be delegated to AI. The majority of physicians refer to tasks such as recommending lifestyle changes based on imaging, generating exam reports, renewing prescriptions only if nothing has changed since the last visit, making a differential diagnosis based on symptoms, recommending therapeutic strategies based on a diagnosis validated by a doctor, HBA1C, and HTN assessment. A few physicians would delegate the evaluation of drug interactions, writing observation sheets and discharge letters, and reporting the evolutions of blood and imaging tests, while others would only delegate something with human interaction (Figure 11).

A total of 93% of physicians from our sample consider AI useful in their daily activities (Figure 12).

One of the main objectives of AI is to improve time management and efficiency in terms of performance, minimize costs, and satisfy customers. The rapid evolution of AI in the health sector created numerous concerns and challenges [3].

Tasks related to establishing appointments, daily activity management, the identification of drug interaction, the registration of discharge letters, the sorting of patients, the early alarm of the deterioration of patients’ conditions, the analysis and interpretation of endoscopic images and videos, managing test results, ECG analysis, issuing medical prescriptions if nothing changed from the last visit, the diagnosis of rare diseases, the assessment of the prognosis of some diseases, the prediction of the effects of therapeutic interventions, therapy planning, the registration of patient history, establishing the diagnosis under an algorithm, and statistical processing for medical protocols assessment, while a small number of physicians would not involve AI. Only one physician mentioned already using AI in their current activity (Figure 13).

Implementing AI within the health sector is driven by the emergence of new technologies and changes in the relationship between physicians and patients. AI helps physicians increase the quality of the services delivered, empowering the patient-centered system [13,14].

The implementation of Artificial Intelligence will be an essential step in medicine. At the moment, some issues still need to be solved. More research is needed in the future to confirm the effectiveness of AI since we observed a need for studies on AI in the health sector from a social responsibility perspective. Understanding the importance of social responsibility can enhance trust among categories of stakeholders. It is essential to understand the impact of AI in medicine through the implementation of new technologies, new platforms, applications, and social networks. Such in-depth research contributes to developing AI technologies and value-added creation [15,16,17,18].

## 5. Conclusions

The current paper presents the attitudes and perceptions of internal physicians regarding AI. We started with an overview of digital transformation. We discussed rules and regulations on artificial intelligence in medicine in Europe and in Romania, including its advantages, benefits, disadvantages, risks, limits, and ethical and socially responsible challenges. After presenting all these aspects, we conducted questionnaires to support our research. 

The involvement of AI in medicine will become a tangible reality, as it will assist physicians in diagnosing and proceeding with surgeries quickly. AI’s main advantage compared to humans is that it can process more data. In the medical profession, it represents a potential tool with many benefits and risks for achieving a specific treatment performance or identifying a disease quickly. 

The limits of AI are represented by the lack of regulation, confidentiality, cybersecurity risks, the dataset size, different types of diseases and new diseases that cannot be easily recognized, and clinical trials that need to be reproduced to successfully launch a drug on the market. 

The current research contributed to a better understanding of AI in healthcare by different stakeholders. Implementing AI and social responsibility will enable professionals to address any challenge. Furthermore, our study extends the perspective of AI by focusing on social responsibility and considering the risks and challenges associated with its implementation. Our results are aligned with those from the literature [14]. The researchers indicated that doctors need training to better understand the advantages, benefits, and risks of AI.

To the best of our knowledge, this research is the first study developed in Romania that investigated the attitudes regarding using artificial intelligence among physicians. Similar to other studies from other countries, the results show the need for physicians to receive training and support in Artificial Intelligence. It also reveals that physicians have positive attitudes, seeing that they, patients, and the medical system will have some advantages and benefits.

AI should not be viewed as a solution to any problem. The human factor is so important and irreplaceable in the health sector that it is essential to check all risks and to check if AI can cause additional issues. The solution could be represented by using AI exclusively under the direct supervision of a physician, like an additional tool, to achieve the best treatment scheme for the patients. 

Measures recommended to be implemented shortly with AI implementation could be represented by

developing educational modules for physicians who address the implementation of AI and the interpretation of data obtainedstimulating training modules focused on informaticsdeveloping and expanding standards that evaluate the performance of AIestablishing regulatory and legal frameworks for AI toolsconducting more research for the development of new methodsconducting controlled trials to test the impact of AI tools on patients, quality of life, costs, and benefitssupporting partnerships between academia, informatics companies, health systems, and community-based practicesincreasing the role of regulations, policies, and procedures on AI in the health sectorincreasing communication to enhance the experience of healthcare professionals and patientsassessing the limits, barriers, and challenges of AI implementation and elaborating strategiesdata management to exclude cyberattacks, data loss, and other risksAI strategies for future potential medical crisespreventive healthcare promotionoptimizing resources and costsunderstanding the patient and the impact that AI has on him

Implementing AI can result in reduced costs due to reduced tests and procedures due to telemedicine platforms that diagnose patients remotely, reducing the cost of hospitalization and speeding up drug discovery, which can help for the better use and allocation of resources. The research concluded that many factors represent barriers to adopting AI. However, organizations need to understand the importance of AI implementation in overcoming the barriers and the disadvantages and taking a strategic advantage. Furthermore, this research provides insights to ensure that future healthcare practitioners are well prepared to implement AI-driven tools. Our study also sheds light on the perceptions of AI in healthcare. AI’s ethical and social responsibility implications in healthcare represent a significant aspect of AI education. The results ensure that future healthcare practitioners are not only open to AI but also sensitive to the socially responsible and ethical dimensions of its use in practice. 

The paper focuses on Artificial Intelligence in medicine. Our findings contribute to the body of knowledge and aim to deepen the understanding of physicians’ perceptions related to AI’s implementation in the healthcare sector. The findings could provide a guideline to higher institutions on artificial intelligence and social responsibility areas to be incorporated into their syllabus. This will ensure that students and future physicians are equipped with the knowledge and skills that meet the current market demands. 

## Figures and Tables

**Figure 1 healthcare-12-01502-f001:**
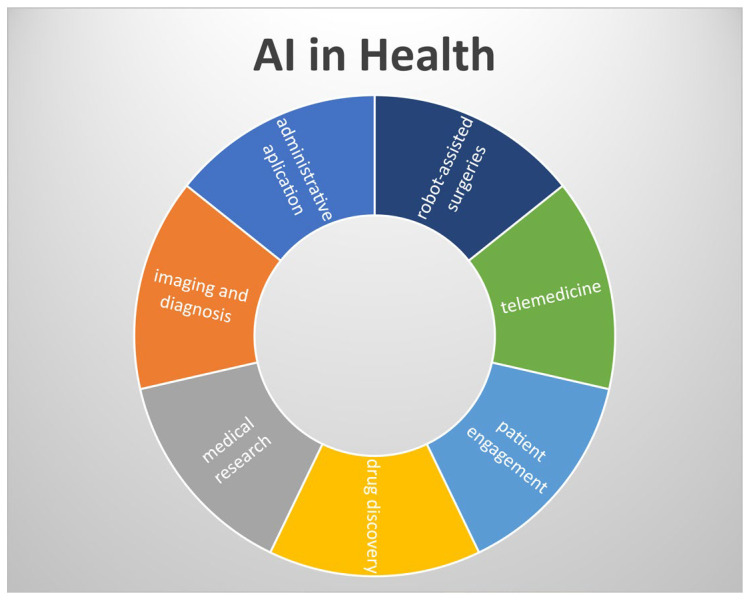
Artificial Intelligence in the health sector.

**Figure 2 healthcare-12-01502-f002:**
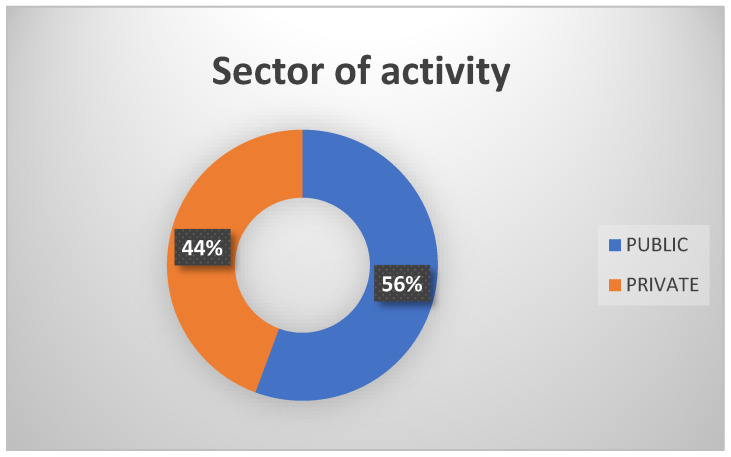
The sector of activity of the study participants.

**Figure 3 healthcare-12-01502-f003:**
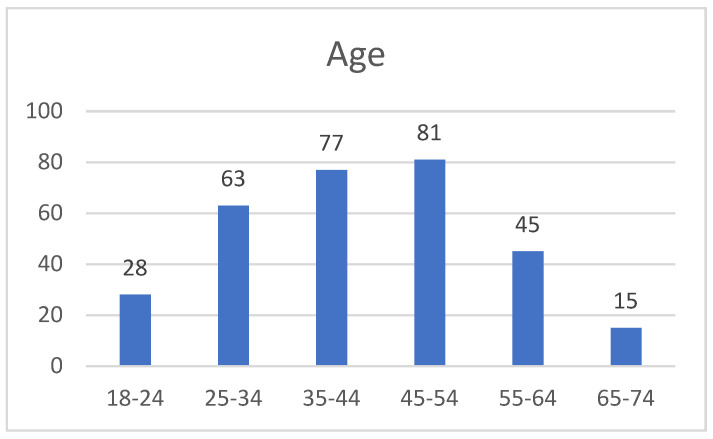
The age distribution of healthcare practitioners that are part of the sample.

**Figure 4 healthcare-12-01502-f004:**
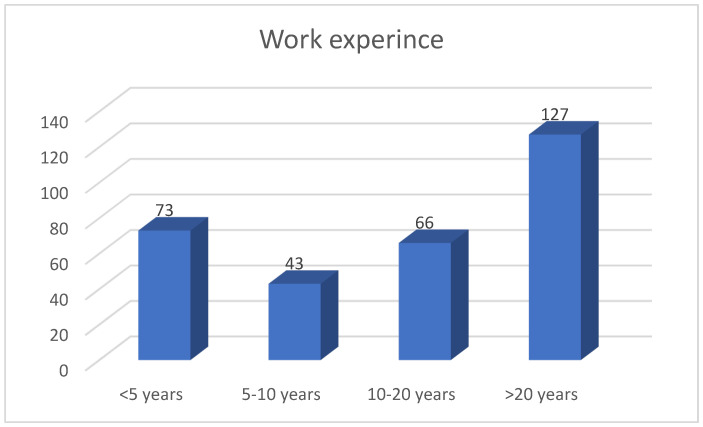
The work experience of the study participants.

**Figure 5 healthcare-12-01502-f005:**
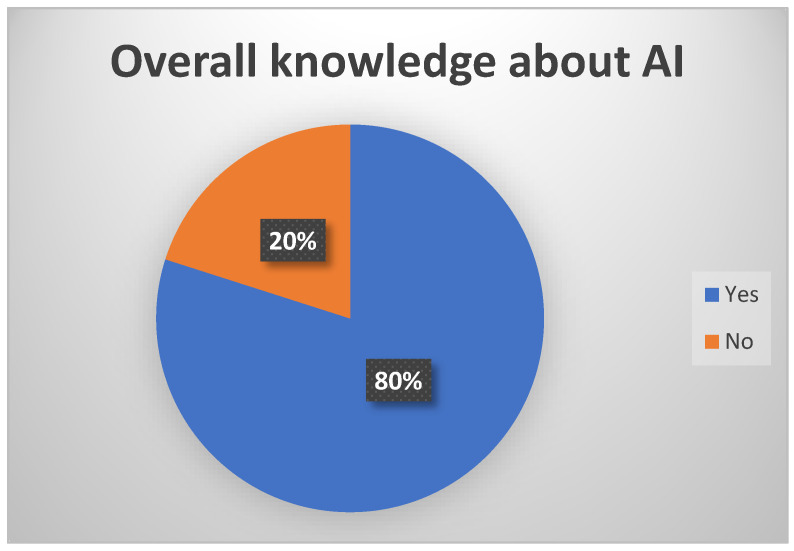
The overall knowledge of AI.

**Figure 6 healthcare-12-01502-f006:**
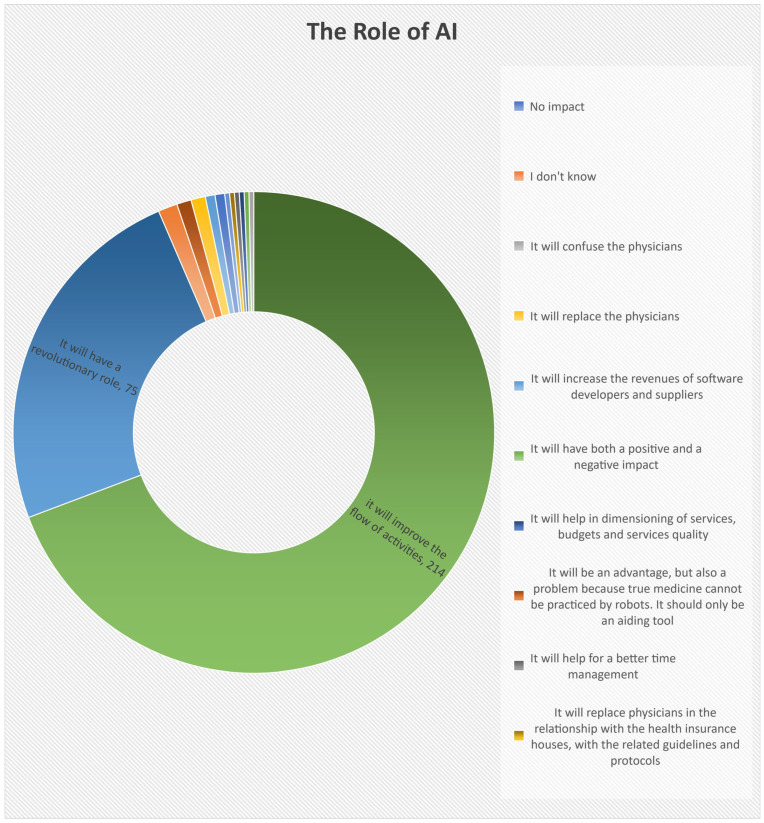
Physicians’ Attitudes Towards the Role of AI.

**Figure 7 healthcare-12-01502-f007:**
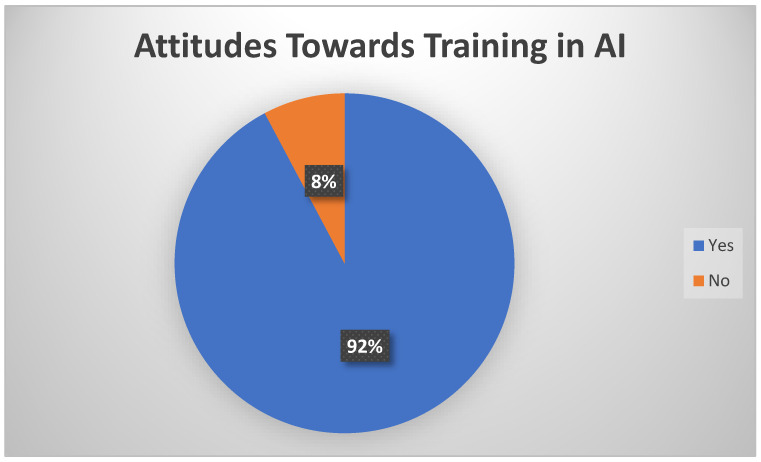
Attitudes toward using artificial intelligence.

**Figure 8 healthcare-12-01502-f008:**
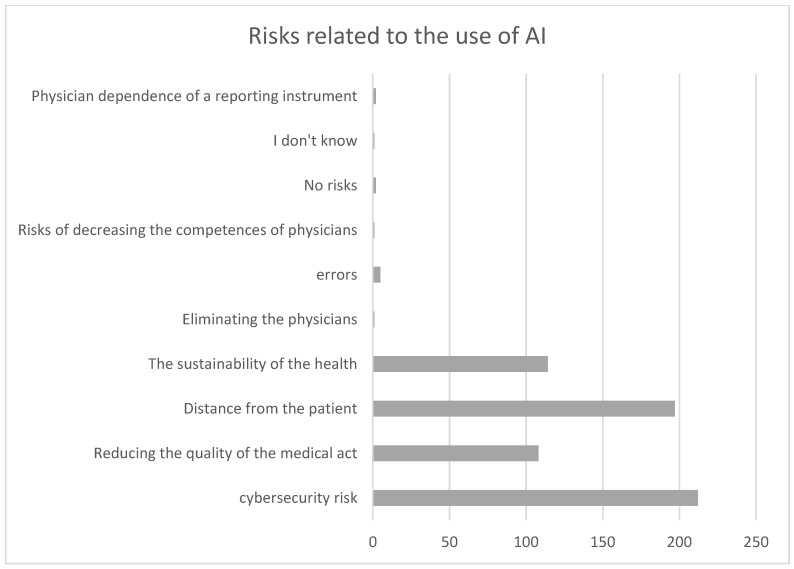
Perceived risks related to the use of AI.

**Figure 9 healthcare-12-01502-f009:**
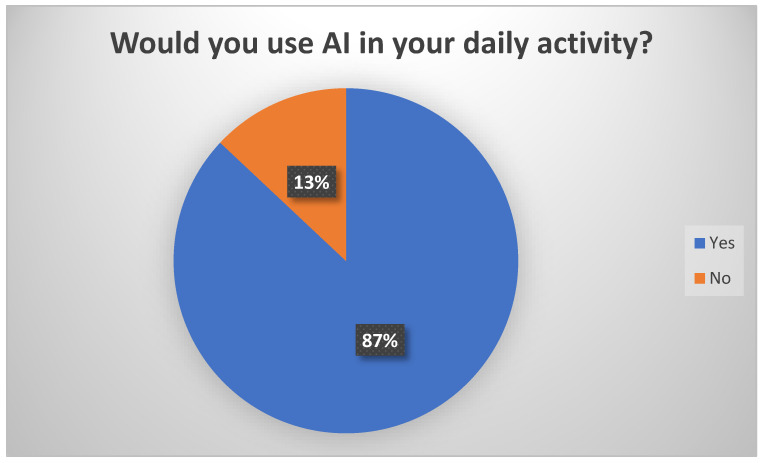
Perceptions of physicians about the use of AI in their daily activity.

**Figure 10 healthcare-12-01502-f010:**
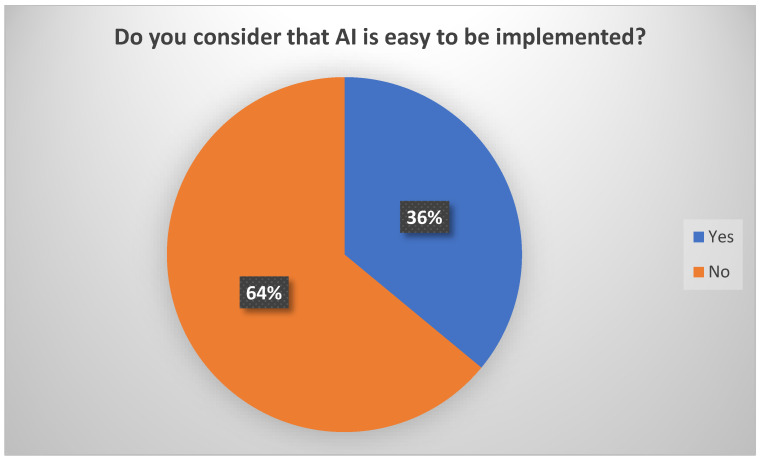
Perceptions of physicians related to AI implementation.

**Figure 11 healthcare-12-01502-f011:**
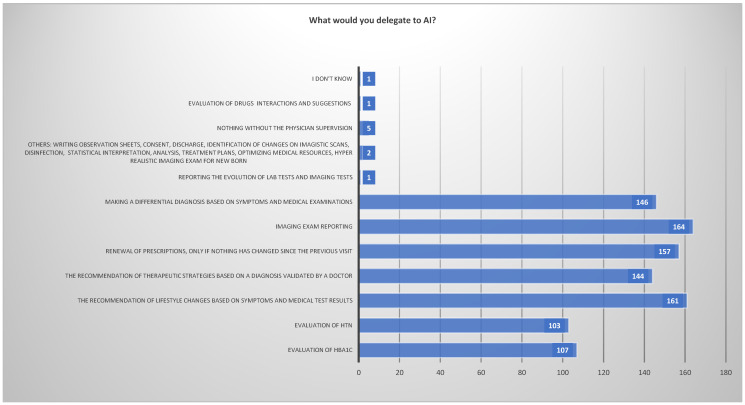
Physicians’ Attitudes towards the Potential Task to be Carried Out by AI.

**Figure 12 healthcare-12-01502-f012:**
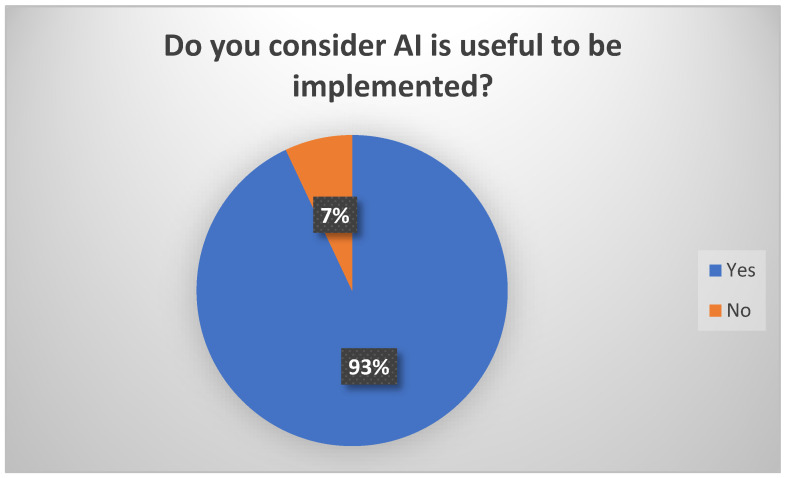
Perceptions related to the usefulness of AI implementations.

**Figure 13 healthcare-12-01502-f013:**
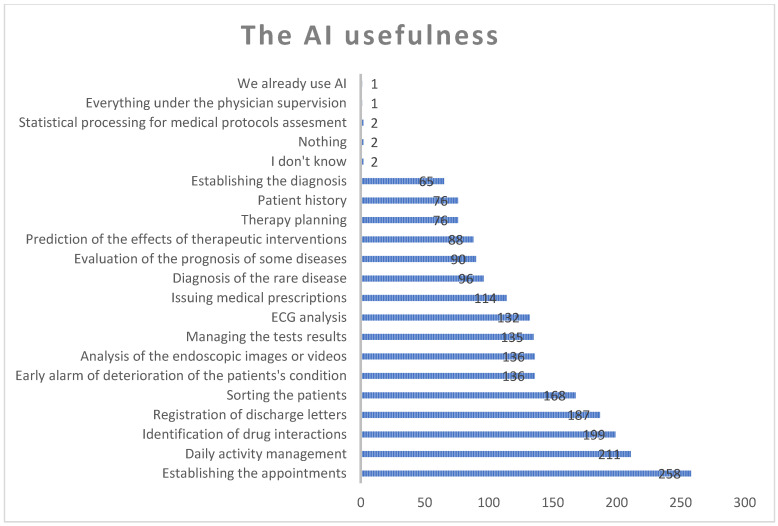
The usefulness of Artificial Intelligence.

**Table 1 healthcare-12-01502-t001:** Study flowchart.

No.	Item	Value (*n*)
1.	Physician selection based on the study protocol	370
2.	Physicians who were excluded from the sample due to no response	60
3.	Physicians excluded from the sample due to missing data	1
4.	Total number of patients	309

**Table 2 healthcare-12-01502-t002:** Questionnaire structure.

Aspects	Questions	Answers
Socio-demographic factors	Q1. What is your age?	Short answer (number)
	Q2. What is your gender?	Female
		Male
	Q3. Which sector of activity are you part of?	Public
		Private
	Q4. What is your professional experience?	Short answer (number)
knowledge	Q5. Do you know about Artificial Intelligence?	Yes
		No
	Q6. Do you think Artificial Intelligence will have a role/impact on the healthcare system?	Yes
		No
	Q7. What do you think its role will be?	It will have a revolutionary role.
		It will improve the flow of activities.
		Other Answers
Attitudes and perceptions	Q8. Do you consider artificial intelligence courses necessary in medical training?	Yes
		No
	Q9. Could artificial intelligence improve the medical act?	Yes
		No
	Q10. What exactly would you delegate to AI?	Evaluation of HbA1C
		Evaluation of HTN
		Recommending lifestyle changes based on symptoms and medical test results
		The recommendation of therapeutic strategies based on a diagnosis validated by a doctor
		Renewal of prescriptions only if nothing has changed since the previous visit
		Imaging exam reporting
		Making a differential diagnosis based on symptoms and medical examinations
		Other Answers
	Q11. Artificial intelligence will	Reduce redundant activities
		It will free physicians from specific tasks
		Others
	Q12. The risks related to the use of artificial intelligence are	Cybersecurity risks
		Reducing the quality of the medical act
		The sustainability of the health system due to the significant investments required to implement this type of technology
		Distance from the patient
		Other Answers
	Q13. Would you include artificial intelligence in your work?	Yes
		No
	Q14. Do you think will it be easy to implement?	Yes
		No
	Q15. Do you think will it be helpful to implement?	Yes
		No
	Q16. It would be helpful for	Analysis of the endoscopic images or videos
		Daily activity management
		Diagnosis of rare diseases
		Early alarm of deterioration of the patient’s condition
		ECG analysis
		Establishing a diagnosis
		Establishing the patients’ appointments
		Evaluation of the prognosis of some diseases
		Identification of drug interactions
		Issuing medical prescriptions if nothing changes since the last visit
		Managing the test results
		Patient history
		Prediction of the effects of therapeutic interventions
		Registration of discharge letters
		Sorting the patients
		The results of the analyses
		Therapy planning
		Other Answers
	Q17. What specialization do you have?	Short answer

**Table 3 healthcare-12-01502-t003:** Healthcare workers’ socio-demographic and professional characteristics. *n* = 309.

Feature	Frequency	Percentage
	Sex	
Male	62	20.06
Female	247	79.94
	Age group	
18–24 years	28	9.06
25–34 years	63	20.39
35–44 years	77	24.92
45–54 years	81	26.21
55–64 years	45	14.56
65–74 years	15	4.86
	Clinical Role	
Physicians	297	96.12
Medical interns and students	11	3.56
Nurses	1	0.32
	Healthcare experience	
<5 years	73	23.62
5–10 years	43	13.92
10–20 years	66	21.36
>20 years	127	41.10
	Sector of activity	
Public	172	55.66
Private	137	44.34

Source: Authors’ projection.

**Table 4 healthcare-12-01502-t004:** The Attitudes of Physicians towards the Role of AI.

Physicians’ Attitudes Towards the Role of AI	Number	Percentage
It will improve the flow of activities.	214	69.27%
It will have a revolutionary role.	75	24.27%
It will have a revolutionary role and it will also improve the flow of activities.	1	0.32%
It will replace physicians in the relationship with the health insurance houses, with the related guidelines and protocols.	1	0.32%
It will help for better time management.	1	0.32%
It will be an advantage but also a problem because robots cannot practice true medicine. It should only be an aiding tool.	3	0.97%
It will help in dimensioning services, budgets, and service quality.	1	0.32%
It will have both a positive and a negative impact.	1	0.32%
It will increase the revenues of software developers and suppliers.	2	0.65%
It will replace physicians.	3	0.97%
It will confuse the physicians.	1	0.32%
I do not know.	4	1.30%
No impact	2	0.65%
Total	309	100%

## Data Availability

Data can be obtained in justified cases from the corresponding author.

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
