# Peer review of "Perceptions of the Impact of Artificial Intelligence among Internal Medicine Physicians as a Step in Social Responsibility Implementation: A Cross-Sectional Study"

_healthcare, 2024, doi:10.3390/healthcare12151502_

Round 1

Reviewer 1 Report

Comments and Suggestions for Authors

Fig 1 must be revised in better look. 

Some paragraphs only consists of one sentence which is not appropriate. Those paragraphs must be extended.

Figure 11. The usefulness of Artificial Intelligence has no meaning, there is no measurement available to understand the bar. It must be revised in meaningful picture and provide in depth discussion accordingly.

Comments on the Quality of English Language

no comment

Author Response

Thank you very much for your time!  

Fig 1 must be revised in better look. 

Response:

Thank you for your suggestion! We decided to remove  Figure 1. Hopefully you will find it to be ok.

Some paragraphs only consists of one sentence which is not appropriate. Those paragraphs must be extended.

Response:

Thank you very much for your comment. Yes, we have restructured the sentences for a better understanding. Hopefully now you will find it rational.

Figure 11. The usefulness of Artificial Intelligence has no meaning, there is no measurement available to understand the bar. It must be revised in meaningful picture and provide in depth discussion accordingly.

Response:

Thank you very much for your suggestion. Yes, you are right! We have taken your comment in consideration and we replaced Figure 11 in revised manuscript with a new figure with better understanding. Hopefully you will find it justified. Sorry for not describing it before. We now provided in depth discussion.

Comments on the Quality of English Language

no comment.

Response:

Thank you very much for your time and valuable feedback! Cordialy Regards!

Reviewer 2 Report

Comments and Suggestions for Authors

Thank you for the opportunity to review this paper. There are a few changes needed to make this paper publishable.

Please find the points to improve the manuscript:

 Abstract:

# The abstract provides a good overview, but try making it more concise. Summarize key findings and implications more precisely.

Introduction:

# The introduction covers the basics of AI well but can be more focused. Specify the unique contributions of this paper early on.

# Ensure consistent use of terminology and concepts throughout the paper. For example, clarify the distinction between AI, machine learning, and deep learning without redundancy.

# Maintain a consistent format for in-text citations and references (e.g., use a standardized style like APA ).

# The flow between sections can be improved. Ensure each paragraph logically follows the previous one, creating a seamless narrative. Clearly define each section's purpose and ensure transitions between sections are smooth.

# Literature Review: Enhance the depth of the literature review by including more recent studies and critical evaluations of existing research. Discuss gaps in current knowledge that your study aims to address.

# Be specific about the socio-demographic factors investigated and their relevance to AI adoption in healthcare. Clearly state the hypotheses or research questions being tested.

Methodology

# The main purpose of the study is mentioned, but it could be more clearly defined. Consider breaking it down into specific, measurable objectives.

# The questionnaire is described briefly, but more details about its development, validation, and reliability would strengthen this section. Describe how the questions were formulated, pre-tested, and revised.

# The sampling method is not clearly explained. Specify whether it was random, convenience, or purposive sampling, and justify the choice of this method.

# The sample size is relatively small. Provide a rationale for this sample size and discuss its implications for the study’s generalizability and statistical power.

# Provide more details about how the survey was administered. Was it online, in-person, or via mail? How were participants recruited? What measures were taken to ensure a high response rate?

# The methodology section should outline the data analysis techniques used. Describe the statistical methods or qualitative analysis techniques applied to interpret the data.

# Strengthen the connection between the methodology and existing literature. Justify your methodological choices by referencing previous studies.

Results and Discussion

# Ensure that each subsection flows logically into the next. For example, the transition from market valuation to physicians' concerns and then to AI applications should be smoother. Clearly separate the discussion of results from the market analysis to avoid confusing the reader.

# Elaborate on the specific concerns raised by physicians regarding job displacement and reliability. Provide more detailed examples or quotes from respondents to strengthen this section. Discuss how these concerns could be addressed practically, possibly integrating more from existing literature or case studies.

# The mention of training needs is frequent but lacks depth. Provide more detailed suggestions on what specific training programs should include and how they could be implemented. Discuss the effectiveness of existing training programs, if any, and how they could be improved based on your findings.

# Expand on the WHO guidelines mentioned. Discuss how these guidelines could be implemented in Romania specifically and what challenges might arise. Compare the WHO guidelines with any national regulations to highlight gaps or alignments.

# The section on attitudes toward AI could benefit from a more nuanced analysis. Discuss any significant differences based on demographics (e.g., age, gender, specialization). Highlight any trends or patterns observed in the attitudes toward AI.

# While listing AI applications, provide more detailed examples and discuss their current state of implementation in Romania. Address any challenges or limitations faced in implementing these applications and potential solutions.

# Expand on the ethical and social responsibility implications. Provide more concrete examples or case studies to illustrate these points. Discuss how ethical considerations could be integrated into AI training programs for healthcare professionals.

# Some points in the results and discussion section seem more appropriate for the conclusion. Consider moving broader statements about the future of AI and its strategic importance to the conclusion. Summarize the key findings and their implications more succinctly in this section, reserving broader reflections for the conclusion.

Wish you all the best!

Comments on the Quality of English Language

Need Minor revisions 

Author Response

Thank you very much for your valuable feedback!

Abstract:

# The abstract provides a good overview, but try making it more concise. Summarize key findings and implications more precisely.

Response:

Thank you very much for your feedback! Yes, we summarized and we presented the implications more precisely in the revised manuscript.

Introduction:

# The introduction covers the basics of AI well but can be more focused. Specify the unique contributions of this paper early on.

Response:

Thank you so much for your observation and valuable comment! Yes, we mentioned the unique contribution of the paper.

# Ensure consistent use of terminology and concepts throughout the paper. For example, clarify the distinction between AI, machine learning, and deep learning without redundancy.

Response:

Thank you very much for your feedback! Sorry for this. We did our best to eliminate redundancy.

# Maintain a consistent format for in-text citations and references (e.g., use a standardized style like APA ).

Response:

Thank you very much! Yes, you are right. We used APA style for both in-text citations and references.

# The flow between sections can be improved. Ensure each paragraph logically follows the previous one, creating a seamless narrative. Clearly define each section's purpose and ensure transitions between sections are smooth.

Response:

Thank you very much for your precious suggestion! Hopefully we achieve your expectations now and we have a mild transition between sections in the revised manuscript.

# Literature Review: Enhance the depth of the literature review by including more recent studies and critical evaluations of existing research. Discuss gaps in current knowledge that your study aims to address.

Response:

Thank you for your comment! We very much appreciate your suggestion and we have updated literature review and we have eliminated the not-so-recent studies, and we have also discussed the gaps that we found.

# Be specific about the socio-demographic factors investigated and their relevance to AI adoption in healthcare. Clearly state the hypotheses or research questions being tested.

Response:

Thank you so much for your comment! Yes, sure, characteristics such as age, gender, specialization, years of professional experience, location, have been asked in the current survey/questionnaire. We mentioned as well the hypotheses of research.

Methodology

# The main purpose of the study is mentioned, but it could be more clearly defined. Consider breaking it down into specific, measurable objectives.

Response:

Thank you for your feedback! We described in a different manner the main purpose of the sudy and we do hope we meet your expectations.

# The questionnaire is described briefly, but more details about its development, validation, and reliability would strengthen this section. Describe how the questions were formulated, pre-tested, and revised.

Response:

Thank you very much for your comment! We presented more details, we described how we tested it and now we do hope that we strengthen this section.

# The sampling method is not clearly explained. Specify whether it was random, convenience, or purposive sampling, and justify the choice of this method.

Response:

Thank you so much for your suggestion! We explained what sampling method we used and we offered all details needed.

# The sample size is relatively small. Provide a rationale for this sample size and discuss its implications for the study’s generalizability and statistical power.

Response:

Thank you very much for your feedback! Yes, totally agree with you, the number of participants is still small (even if now we have 309 participants in our study) and the reduced sample size can limit the generalizability of the results obtained. Despite the small sample size, we feel now that the participants work in our favor, in terms of being able to generalize what we found to a larger sample.

# Provide more details about how the survey was administered. Was it online, in-person, or via mail? How were participants recruited? What measures were taken to ensure a high response rate?

Response:

Thank you so much for your suggestion! Yes, sure! The main purpose of the study was to focus on AI in health sector, and to examine the perceptions of internal medicine physicians on AI implementation in healthcare sector, on main challenges, and main risks, in accordance with social responsibility. A cross-sectional survey was performed between February-June 2024 among 309 physicians from Romania, from both the public and private sectors. The participant selection process took into account the specialization of the physicians. Each participant provided informed and voluntary consent, before completing the survey, under the ethical guidelines under the Declaration of Helsinki, accomplishing transparency and ethical integrity of the current study. The number of healthcare professionals from Romania, according to the last published data from The National Institute of Statistics, is 30. 639 are physicians, 21.430 are stomatologists, 22.660 are pharmacists, and 18.910 nurses. An electronic survey was proposed during February - June 2024 to a sample of healthcare professionals, 309 were admitted to the study, of which 62 males, 247females, mean age 42 years. Data was gathered through an in-depth questionnaire across hospitals and clinics in Romania from 309 health professionals that enriched the understanding of how much the need is for social responsibility actions and activities when integrating AI into medicine. The number of physicians included in the sample was 370, of which 60 were excluded because they did not complete the survey and another one due to the missing data in the completed survey. The findings show a continuous intention to use AI despite the lack of regulation on both AI and social responsibility in the health sector. We used open-ended questions, true or false, and multiple-choice questions to explore both concepts. A total of 309 questionnaires were collected in different rounds (Table 1)

No.

Item

Value (n)

1.

Physician selection based on the study protocol

370

2.

Physicians who were excluded from the sample due to no response

60

3.

Physicians excluded from the sample due to missing data

1

4.

Total number of patients

309

Table 1 Study flowchart

The first stage of the study was exploratory questionnaires tested on a small sample (N=10). In the second stage, starting from the answers received, we reevaluated and rephrased the questions. We used non-random or convenience sampling since it was easy to. access to data (geographical proximity, availability at a given time, and willingness to participate in the research). Another reason why we used convenience sampling is that is often used in qualitative and medical research studies when involves selecting participants that are available around a particular location. We are researching physician perceptions of AI implementation in healthcare, in the city of Bucharest, Romania. We have determined that a sample of 309 physicians is sufficient to answer our research question and to discover the physicians’ perceptions, attitudes, and opinions on AI. The study can be developed in future research paths. To collect our data, we sent messages through a pre-existing group of internal medicine physicians and asked them if they wanted

 to participate in our research and to complete our online survey. To maximize the response ratio, we continued to invite them to complete the survey and send them reminders at different days and times, during February – June 2024, with the shortened URL link, until the sample size was reached. We mentioned that we did not offer monetary or other incentives to complete our survey. As a method of research, we used the descriptive analysis method. The limit of our research is represented by the fact that it cannot be generalized to the target population.

# The methodology section should outline the data analysis techniques used. Describe the statistical methods or qualitative analysis techniques applied to interpret the data.

Response:

Thank you for your feedback! Within the previous comment, we presented all the desired details.

# Strengthen the connection between the methodology and existing literature. Justify your methodological choices by referencing previous studies.

Response:

Thank you for your precious advice! We solved also this aspect.

Results and Discussion

Ensure that each subsection flows logically into the next. For example, the transition from market valuation to physicians' concerns and then to AI applications should be smoother. Clearly separate the discussion of results from the market analysis to avoid confusing the reader.

Response:

Thank you very much for your suggestion! Hopefully, we achieve your expectations now and we have a mild transition.

# Elaborate on the specific concerns raised by physicians regarding job displacement and reliability. Provide more detailed examples or quotes from respondents to strengthen this section. Discuss how these concerns could be addressed practically, possibly integrating more from existing literature or case studies.

Response:

Thank you so much for your suggestion! Yes, we offered quotes from participants of the study related to job displacement, cybersecurity concerns, mal parxis concerns, and other aspects raised by physicians.

# The mention of training needs is frequent but lacks depth. Provide more detailed suggestions on what specific training programs should include and how they could be implemented. Discuss the effectiveness of existing training programs, if any, and how they could be improved based on your findings.

Response:

Thank you so much for your comment! Yes there is a need for training in AI, seeing that there are differences in the answers received. We observed that physicians with more years of experience are more reticent to AI use in their daily activities compared to juniors. Specific training programs should include and should not be limited to Introduction to Responsible AI; AI for Physicians AI, Medicine and the Future of Work; Python, Natural Language Processing, Advanced Generative AI, DeepFake, and Voice Cloning.

# Expand on the WHO guidelines mentioned. Discuss how these guidelines could be implemented in Romania specifically and what challenges might arise. Compare the WHO guidelines with any national regulations to highlight gaps or alignments.

Response:

Thank you very much for your feedback! We discussed the regulations at the European Union level to compare with the situation in Romania.

# The section on attitudes toward AI could benefit from a more nuanced analysis. Discuss any significant differences based on demographics (e.g., age, gender, specialization). Highlight any trends or patterns observed in the attitudes toward AI.

Response:

Thank you very much for your advice!

# While listing AI applications, provide more detailed examples and discuss their current state of implementation in Romania. Address any challenges or limitations faced in implementing these applications and potential solutions.

Response:

Thank you very much for your feedback! Sure, we discussed the situation from Romania and also the challenges we face in the implementation process.

# Expand on the ethical and social responsibility implications. Provide more concrete examples or case studies to illustrate these points. Discuss how ethical considerations could be integrated into AI training programs for healthcare professionals.

Response:

Thank you for your sugesstions! Yes we mentioned the ethical considerations.

# Some points in the results and discussion section seem more appropriate for the conclusion. Consider moving broader statements about the future of AI and its strategic importance to the conclusion. Summarize the key findings and their implications more succinctly in this section, reserving broader reflections for the conclusion.

Response:

Thank you for your feedback! We solved these aspects.  

Wish you all the best!

Comments on the Quality of English Language

Need Minor revisions 

Response:

Thank you for your recommendation! We made English revisions, to address minor editing of English language as required.

Response:

Thank you very much for your time and precious feedback! Cordialy Regards!

Reviewer 3 Report

Comments and Suggestions for Authors

Dear Editor, 

Thank you for the chance to review the manuscript "Artificial Intelligence in Healthcare, a Step for Social Responsibility Implementation" by Constatntin & Dumitrascu.

The manuscript has a potential to be of interest to the readers, but I have several major remarks:

1. It seems as a fusion between review and an original research article with the Sections 2 & 3 that overview AI in healthcare and then Section 4 & 5 than present results from a survey conducted by the authors. The manuscript lacks clarity and coherences as it namely tries to combine two different types of manuscripts. I would suggest the authors divide the manuscript in two separate manuscripts: review on AI in healthcare and original article that presents the conducted survey. 

2. My major concern is directed at the survey: First, 29 respondents  is too low count for such a socially important question. Moreover, the authors did not present the methodology clearly: how were the physicians selected, how were their contact details obtained,  did the survey undergo an ethical approval, did the respondents consent in any way in participating, what was the target number, out of how many reached, how many did respond. Those are all relevant questions in qualitative research that the authors failed to mention. Because of the gap in methodology description I assume that the authors have tried to circumvent it by fusing the research article with review one. Importantly, the authors did not provide the questionnaire that was used and thus the reader is not able to comprehend the utility of the questionnaire. Providing the questionnaire can only be beneficial to the authors as their questionnaire may be subsequently validated by others. My suggestion to the authors is to make efforts and increase the respondents number and also present the methodology in a rigorous way.

3. There is too much repetition in the text.

Comments on the Quality of English Language

Minor editing may be required

Author Response

Thank you!

The manuscript has a potential to be of interest to the readers, but I have several major remarks:

  1. It seems as a fusion between review and an original research article with the Sections 2 & 3 that overview AI in healthcare and then Section 4 & 5 than present results from a survey conducted by the authors. The manuscript lacks clarity and coherences as it namely tries to combine two different types of manuscripts. I would suggest the authors divide the manuscript in two separate manuscripts: review on AI in healthcare and original article that presents the conducted survey. 

Response:

Thank you very much for your comment! We redesigned these sections. Hopefully now you find them more logical.

  1. My major concern is directed at the survey: First, 29 respondents is too low count for such a socially important question. Moreover, the authors did not present the methodology clearly: how were the physicians selected, how were their contact details obtained,  did the survey undergo ethical approval, did the respondents consent in any way to participating, what was the target number, out of how many reached, how many did respond. Those are all relevant questions in qualitative research that the authors failed to mention. Because of the gap in methodology de

scription, I assume that the authors have tried to circumvent it by fusing the research article with review one. Importantly, the authors did not provide the questionnaire that was used and thus the reader is not able to comprehend the utility of the questionnaire. Providing the questionnaire can only be beneficial to the authors as their questionnaire may be subsequently validated by others. My suggestion to the authors is to make efforts and increase the respondents number and also present the methodology in a ri.gorous way..

Response:

Thank you so much for your suggestion! Yes, agree! We increased the number of participants of the study. We also provided the ethical committee approval no. 2 from January 3rd, 2024.

The main purpose of the study was to focus on AI in the health sector and to examine the perceptions of internal medicine physicians on AI implementation in the healthcare sector, on main challenges, and main risks, under social responsibility. A cross-sectional survey was performed between February and June 62024 among 309 physicians from Romania, from both the public and private sectors. The participant selection process took into account the specialization of the physicians. Each participant provided informed and voluntary consent, before completing the survey, under the ethical guidelines under the Declaration of Helsinki, accomplishing transparency and ethical integrity of the current study. The number of healthcare professionals from Romania, according to the last published data from The National Institute of Statistics, is 30.639 are physicians, 21.430 are stomatologists, 22.660 are pharmacists, and 18.910 nurses. An electronic survey was proposed during February - June 2024 to a sample of healthcare professionals, 309 were admitted to the study, of which 62 males, and 247 females, a mean age of 42 years. Data was gathered through an in-depth questionnaire across hospitals and clinics in Romania from 309 health professionals that enriched the understanding of how much the need is for social responsibility actions and activities when integrating AI into medicine. The number of physicians included in the sample was 370, of which 60 were excluded because they still needed to complete the survey and another one due to the missing data in the completed survey. The findings show a continuous intention to use AI despite the lack of regulation on both AI and social responsibility in the health sector. We used open-ended questions, true or false, and multiple-choice questions to explore both concepts. A total of 309 questionnaires were collected in different rounds (Table 1)

No.

Item

Value (n)

1.

Physician selection based on the study protocol

370

2.

Physicians who were excluded from the sample due to no response

60

3.

Physicians excluded from the sample due to missing data

1

4.

Total number of patients

309

Table 1 Study flowchart

The first stage of the study was exploratory questionnaires, tested on a small sample (N=10). In the second stage, starting from the answers received, we reevaluated and rephrased the questions. We used non-random or convenience sampling since it was easy to have access to data (geographical proximity, availability at a given time, willingness to participate in the research). Another reason why we used convenience sampling is that is often used in qualitative and medical research studies when involves selecting participants that are available around a particular location. We are researching physician perceptions of AI implementation in healthcare, in the city of Bucharest, Romania. We have determined that a sample of 309 physicians is sufficient to answer our research question and to discover the physicians’ perceptions, attitudes, and opinions on AI. The study can be developed in future research paths. To collect our data, we sent messages through a pre-existing group of internal medicine physicians and asked them if they wanted to participate in our research and to complete our online survey. To maximize the response ratio, we continued to invite them to complete the survey and send them reminders at different days and times, during February – June 2024, with the shortened URL link, until the sample size was reached. We mentioned that we did not offer monetary or other incentives to complete our survey. As a method of research, we used the descriptive analysis method. The 0limit of our research is represented by the fact that it cannot be generalized to the target population.

  1. There is too much repetition in the text.

Response:

Thank you very much for your feedback! We solved this aspect. Hopefully, now you find it more logical.

Comments on the Quality of English Language

Minor editing may be re-required.

Response:

Thank you for your recommendation! We made English revisions, to address minor editing of the

English language as required.

Thank you very much for your time and precious feedback! Cordialy Regards!

Reviewer 4 Report

Comments and Suggestions for Authors

The paper is interesting and within the scope of the journal. It is also generally well written, and has potential to reach required quality standards. However in my view the paper is not acceptable in its current form and needs to be redesigned to be acceptable. AI in healthcare is nowadays too broad topic to be highlighted as the main perspective, the paper needs to be more focused on the social study conducted. Also, for such a wide title, authors might need to embrace some coauthor with strong publication record and proven competence in the field.

Here are the suggestions to the authors:

* Consider redesigning the title and the paper in order to make it more focused on the social study performed.

* Number of the subjects (29) might be too low for the wide perspective, especially since the survey was conducted as an electronic one. Please try to include more subjects.

* Figures need to be referenced in the caption also, not only in text (e.g. Figure 1, taken from [1], ref [1] should also be referenced in caption)

* Some figures do not bring any merit to the paper, such as Figure 1.

* References list needs to be improved. For example [1] is too old to be leading ref regarding AI nowadays. Ref [2] is not proper citation of Internet source., etc.

* Novelty of the paper has to be more clearly stated in the concluding paragraph of the Introduction. Reviewing the literature regarding AI in healthcare is too ambitious goal, and needs to be better reflected in the list of authors.

* Proclaimed goal of the paper in the last paragraph of Introduction is not fully in line with title, abstract and contents. But it should anyway be improved as suggested in the previous remark.

* In Figure 5 low number of participants in the study appears critical when divided in the specializations. Even with increased number of participants this figure might still not be advisable. This essentially shows the main problem regarding wide scope of the paper and moderate number of participants, even if the paper is more focused to social study.

* Results and Discussions section needs to be more focused, in compliance with change of title and paper perspective.

* Conclusions section should be somewhat more concise and in line with improved paper structure suggested above.

Still, with moderate editing, increased number of participants of the study and change of focus paper might be transformed to acceptable form, which I am looking forward to.

Comments on the Quality of English Language

The quality of English is generally OK, text is well written and readable. Only minor editing might be needed, but that might be an issue only when and if the paper reaches final publication stage, but in my view for that significant changes are needed. Quality of English is currently the secondary issue.

Author Response

The paper is interesting and within the scope of the journal. It is also generally well written, and has potential to reach required quality standards. However in my view the paper is not acceptable in its current form and needs to be redesigned to be acceptable. AI in healthcare is nowadays too broad topic to be highlighted as the main perspective, the paper0 needs to be more focused on the social study conducted. Also, for such a wide title, authors might need to embrace some coauthor with .strong publication record and proven competence in the field.

Response:

Here are the suggestions to the authors:

* Consider redesigning the title and the paper to make it more focused on the social study performed.

Thank you very much for your suggestion! We have taken your comment into3. full consideration and we redesigned the title of the paper. 

Response:

* Number of the subjects (29) might be too low for the wide perspective, especially since the survey was conducted as an electronic one. Please try to include more subjects.

Thank you very much for your precious advice! Yes, we agree. We extended our sample and we explained everything in the Methodology Section.

Response:

* Figures need to be referenced in the caption also, not only in text (e.g. Figure 1, taken from [1], ref [1] should also be referenced in the caption)

Thank you so much for your comment! Sure, we agree. Referring to Figure 1, we eliminated it from the paper, because does not bring any merit to the paper, as you mentioned.

Response:

* Some figures do not bring any merit to the paper, such as Figure 1.

Thank you so much for your observation! Sure, we agree. For that reason we eliminated this Figure. 

Response:

* References list needs to be improved. For example [1] is too old to be leading ref regarding AI nowadays. Ref [2] is not proper citation of Internet source., etc.

Thank you very much! Yes, you are right, we revised the references. We used APA style for both in-text citations and references

Response:

* Novelty of the paper has to be more clearly stated in the concluding paragraph of the Introduction. Reviewing the literature regarding AI in healthcare is too ambitious goal, and needs to be better reflected in the list of authors.

Thank you very much for your comment! Yes, we redesigned the Introduction section.

Response:

* Proclaimed goal of the paper in the last paragraph of Introduction is not fully in line with title, abstract and contents. But it should anyway be improved as suggested in the previous remark.

Thank you so much for your suggestion! Yes, we redesigned the Introduction section, as well.

Response:

* In Figure 5 low number of participants in the study appears critical when divided into specializations. Even with an increased number of participants, this figure might still not be advisable. This essentially shows the main problem regarding the wide scope of the paper and a

moderate number of participants, even if the paper is more focused on social study.

Thank you for your feedback! Yes, but now we redesigned our study and hopefully you find it ok.

Response:

* Results and Discussions section needs to be more focused, in compliance with change of title and paper perspective.

Thank you for your feedback! Yes, we restructured the article and redesign it, and this section is more focused and in compliance with the title that was also revised.

Response:

* Conclusions section should be somewhat more concise and in line with improved paper structure suggested above.

Response:

For better interpretation, the conclusion section is revised. Thank you very much for your recommendation!

Still, with moderate editing, increased number of participants of the study and change of focus paper might be transformed to acceptable form, which I am looking forward to.

Response:

Thank you so much for your precious feedback! Yes, you are right. We increased the number of participants and we have now a new version of the study.   

Comments on the Quality of English Language

The quality of English is generally OK, text is well written and readable. Only minor editing might be needed, but that might be an issue only when and if the paper reaches final publication stage, but in my view for that significant changes are needed. Quality of English is currently the secondary issue.

Response:

Thank you for your feedback! We edited the document. Hopefully you will find it ok now.

Thank you very much for your time and valuable feedback! Cordialy Regards!

Reviewer 5 Report

Comments and Suggestions for Authors

The paper examines the implementation of artificial intelligence (AI) in healthcare, highlighting its market growth, benefits, challenges and the perspectives of professionals. It discusses the regulatory guidelines, industry development, and various AI applications in healthcare. The author emphasizes the importance of training and cooperation between healthcare institutions and technology developers. 

Major comments
Add details on the quantitative data supporting the estimated market growth such as annual growth rate, market share percentage or any specific monetary value. This will be better illustration to show the market growth over a period.

Methods: Elaborate on how the data was collected and analyzed. How were the participants selected and what was the criteria for participation? 

Minor comments
The review can benefit from more in-depth analysis of specific AI tools and their current real-world applications. Some statistical analysis will be needed to show the significance of the data in the respective figures. 

The manuscript is clear and relevant for the field, and addresses current trends and future directions of AI in healthcare. The manuscript is scientifically reasonable, details on the methods will help to add better understand the reproducibility of the results. The conclusion is consistent with the evidence and argument. The manuscript can be considered after the major comments are addressed. 

Author Response

Thank you very much for your time!

The paper examines the implementation of artificial intelligence (AI) in healthcare, highlighting its market growth, benefits, challenges and the perspectives of professionals. It discusses the regulatory guidelines, industry development, and various AI applications in healthcare. The author emphasizes the importance of training and cooperation between healthcare institutions and technology developers. 

Major comments
Add details on the quantitative data supporting the estimated market growth such as annual growth rate, market share percentage or any specific monetary value. This will be better illustration to show the market growth over a period.

Response:

Thank you for your feedback! We redesigned the manuscript

Methods: Elaborate on how the data was collected and analyzed. How were the participants selected and what was the criteria for participation? 

Response:

Thank you for your feedback! We mentioned this within the Methodology section.

Minor comments
The review can benefit from a more in-depth analysis of specific AI tools and their current real-world applications. Some statistical analysis will be needed to show the significance of the data in the respective figures. 

Response:

Thank you very much for your suggestions!

The manuscript is clear and relevant for the field, and addresses current trends and future directions of AI in healthcare. The manuscript is scientifically reasonable, details on the methods will help to add better understand the reproducibility of the results. The conclusion is consistent with the evidence and argument. The manuscript can be considered after the major comments are addressed. 
Response:

Thank you for your suggestions! We edited the document. Hopefully you will find it ok.

Thank you very much for your time and valuable feedback!

Cordialy Regards!

Round 2

Reviewer 4 Report

Comments and Suggestions for Authors

The paper remains interesting and within the scope of the journal, and remains generally well written. Authors have invested considerable effort into editing paper and transformed it greatly. I believe that quality of the paper has been significantly improved, and I recognize the effort to address and embrace all of my remarks. 

Since paper has been edited very heavily, I believe that another round of careful reading and editing is needed, since it is now very hard to read heavily edited version with markings. By editing the title, shifting the focus and increasing the number of the participants in the research study major issues have been resolved. Still, some concerns remain:

* Title of the paper is now much better regarding shifted focus of the paper, but it might need final editing for clarity.

* Significant increase of the participants in the study in a very short period of time demands comment from authors - how has this been accomplished?

* I still feel that novelty of the paper expressed in the last paragraphs of introduction section has to be edited, expressed more clearly and adjusted to novel narrower focus of the paper.

* Conclusions section might need further improvements t better reflect novel structure of the paper.

 I strongly encourage authors to perform another round of editing since much improved paper just need some final adjustments to reach required quality and form.

Comments on the Quality of English Language

English still seems OK, but due to too many edits it is now difficult to access readability of the paper. Final proofreading might be needed.

Author Response

The paper remains interesting and within the scope of the journal, and remains generally well written. Authors have invested considerable effort into editing paper and transformed it greatly. I believe that quality of the paper has been significantly improved, and I recognize the effort to address and embrace all of my remarks. 

Since paper has been edited very heavily, I believe that another round of careful reading and editing is needed, since it is now very hard to read heavily edited version with markings. By editing the title, shifting the focus and increasing the number of the participants in the research study major issues have been resolved. Still, some concerns remain:

Response: Thank you very much for your feedback! We very much appreciate all your comments and constructive suggestions. They were very important to us and helped to improve our work. We strongly believe that these increased the value of our manuscript. We present below the responses to all your comments:

* Title of the paper is now much better regarding shifted focus of the paper, but it might need final editing for clarity.

Response: Thank you very much for your recommendation! Yes, we carefully read, checked, and edited the document. Due to the use of Track Changes, some errors occurred. We are sorry for the inconvenience. Hopefully, now you will find it ok.

* Significant increase of the participants in the study in a very short period of time demands comment from authors - how has this been accomplished?

Response: Thank you very much for your comment! Yes, you are right! As suggested, we did a teamwork-intensive effort and we reviewed the article, together with our co-authors, and asked physicians to complete our survey. We were reaching out to our respondents through different channels, sending the link through email, sending them gentle reminders to complete the survey, or WhatsApp messages also as reminders. We used the social media platform where they were more active and ready to help us with their answers. Another important aspect is timing. We avoided posting it during busy times when they were maybe less likely to respond. The main purpose of the study was to focus on AI in the health sector and to examine the perceptions of internal medicine physicians on AI implementation in the healthcare sector, on main challenges, and main risks, in accordance with social responsibility. A cross-sectional survey was performed from February to June 2024 among 309 physicians from Romania, from both the public and private sectors. The participant selection process took into account the specialization of the physicians. Each participant provided informed and voluntary consent, before completing the survey, under the ethical guidelines under the Declaration of Helsinki, accomplishing transparency and ethical integrity of the current study. The number of healthcare professionals from Romania, according to the last published data from The National Institute of Statistics, is 71. 279 are physicians, 21.430 are stomatologists, 22.660 are pharmacists, and 18.910 nurses. An electronic survey was proposed during February - June 2024 to a sample of healthcare professionals, 309 were admitted to the study, of which 62 were males, and 247 were females, mean age of 42 years. Data was gathered through an in-depth questionnaire across hospitals and clinics in Romania from 309 health professionals that enriched the understanding of how much the need is for social responsibility actions and activities when integrating AI into medicine. The number of physicians included in the sample was 370, of which 60 were excluded because they did not complete the survey and another one due to the missing data in the completed survey (he did not mention the age). The findings show a continuous intention to use AI despite the lack of regulation on both AI and social responsibility in the health sector. We used open-ended questions, true or false, and multiple-choice questions to explore both concepts. A total of 309 questionnaires were collected in different rounds (Table 1)

No.

Item

Value (n)

1.

Physician selection based on the study protocol

370

2.

Physicians excluded from the sample due to no response

60

3.

Physician excluded from the sample due to missing data

1

4.

Total number of patients

309

                         Table 1 Study flowchart

The first stage of the study was exploratory questionnaires, tested on a small sample (N=10). In the second stage, starting from the answers received, we reevaluated and rephrased the questions. We used non-random or convenience sampling because it was easy to have access to data (geographical proximity, availability at a given time, and willingness to participate in the research). Another reason why we used convenience sampling is that is often used in qualitative and medical research studies when involves selecting participants that are available around a particular location. We are researching physician perceptions of AI implementation in healthcare, in the city of Bucharest, Romania. We have determined that a sample of 309 physicians is sufficient to answer our research question and to discover the physicians’ perceptions, attitudes, and opinions on AI. The study can be developed in future research paths. To collect our data, we sent messages through a pre-existing group of internal medicine physicians and asked them if they wanted to participate in our research and to complete our online survey. To maximize the response ratio, we continued to invite them to complete the survey and send them reminders at different days and times, during February – June 2024, with the shortened URL link, until the sample size was reached. As a method of research, we used the descriptive analysis method. The limit of our research is represented by the fact that it cannot be generalized to the target population.

* I still feel that novelty of the paper expressed in the last paragraphs of introduction section has to be edited, expressed more clearly and adjusted to novel narrower focus of the paper.

Response: Thank you very much for your comment! Yes, you are right. The paper focuses on Artificial Intelligence in medicine. Our findings contribute to the body of knowledge and it aims to deepen the understanding of perceptions of physicians related to AI’s implementation in the healthcare sector. The findings could provide a guideline to higher institutions on artificial intelligence and social responsibility areas to be incorporated into their syllabus. This will ensure that students, and future physicians, are equipped with knowledge and skills that meet the nowadays market demands. To the best of the authors’ knowledge, this is the first study to investigate the physicians’ perspectives on artificial intelligence in our country.   

* Conclusions section might need further improvements to better reflect novel structure of the paper.

Response:   Thank you for your feedback! For better interpretation, we revised the conclusion section, according to the new structure of the paper.

* I strongly encourage authors to perform another round of editing since much improved paper just need some final adjustments to reach required quality and form.

Response: Many thanks for your suggestion! Yes, we read again the paper and you are right. We edited the entire document. We made English revisions, to address minor editing of the English language as required. We are sorry for the inconvenience. We made the adjustments, and now hopefully the quality and form of the paper meet your expectations.

Comments on the Quality of English Language

English still seems OK, but due to too many edits it is now difficult to access readability of the paper. Final proofreading might be needed.

Response: Many thanks for your comment! Yes, we read again the paper and you are right. We have considered your comment and we revised the manuscript, to check the English proficiency. Sorry for this! We did our best to eliminate any mistake that occurred. Hopefully, now is easy to access the readability of the paper.

Submission Date

22 May 2024

Date of this review

10 Jul 2024 00:20:23

Please address all correspondence concerning this manuscript to me at mihaela.dumitrascu@cig.ase.ro.

Thank you very much for your valuable suggestions! We agree with you and we appreciate very much your time and dedication to this manuscript review!

Cordial Regards,

Luminita-Mihaela DUMITRASCU,

On behalf of all co-authors

14.07.2024
